# Gender Differences in Dual Diagnoses Associated with Cannabis Use: A Review

**DOI:** 10.3390/brainsci12030388

**Published:** 2022-03-15

**Authors:** Laura Prieto-Arenas, Ignacio Díaz, M. Carmen Arenas

**Affiliations:** 1Facultad de Medicina, Universidad Católica de Valencia San Vicente Mártir, 46001 Valencia, Spain; laurapriar@gmail.com (L.P.-A.); nachodfa@gmail.com (I.D.); 2Unidad de Salud Mental de Xàtiva, 46800 Valencia, Spain; 3Facultad de Psicología y Logopedia, Universitat de València, 46010 Valencia, Spain

**Keywords:** cannabis, gender differences, psychotic disorders, depression, anxiety, male, female, humans

## Abstract

Gender differences in psychiatric disorders and drug use are well known. Cannabis is the most widely used illegal drug among young people. In recent years, its use has been related to the development of psychiatric pathologies; however, few studies have incorporated the gender perspective as of yet. The present work analyses the literature to determine the existence of gender differences in the development of psychotic, depressive and anxious symptoms associated with cannabis use. First, we describe cannabis misuse and its consequences, paying special attention to adolescent subjects. Second, the main gender differences in psychiatric disorders, such as psychosis, depression, anxiety and cannabis use disorders, are enumerated. Subsequently, we discuss the studies that have evaluated gender differences in the association between cannabis use and the appearance of psychotic, depressive and anxious symptoms; moreover, we consider the possible explanations for the identified gender differences. In conclusion, the studies referred to in this review reveal the existence of gender differences in psychiatric symptoms associated with cannabis use, although the direction of such differences is not always clear. Future research is necessary to discern the causal relationship between cannabis use and the development of psychiatric symptoms, as well as the gender differences found.

## 1. Introduction

Men and women differ in the prevalence and manifestation of many psychiatric disorders [1,2]. At present, the higher prevalence of psychosis in men is well established [3,4,5], whereas women manifest depressive and anxiety disorders to a greater extent [6,7,8,9,10]. Similarly, there are many differences between men and women in addictive disorders [11,12,13,14,15].

Men have long been the subject of research on drug use, given that men typically have a higher prevalence of drug use. However, this fact differs from the current reality. The latest statistics show that women are catching up with men in drug use, especially in the younger population [16,17]. Broadly speaking, it can be observed that women have a higher prevalence in the use of legal drugs such as tobacco and alcohol, as well as in the use of prescription and non-prescription hypnosedatives. In contrast, men continue to be the more prevalent users of illegal drugs such as cannabis, psychostimulants, hallucinogens and opiates. Although women do not consume as much cannabis as men, an increase can be observed in the latest reports with respect to the statistics from previous years [16,17]. Hence, the gender gap in the prevalence of drug use is already starting to narrow.

Cannabis has been the most widely consumed illegal drug in recent years, with an upward trend mainly among young people [18]. The legalization of this drug in some countries, as well as the development of its therapeutic use, has led to an underestimation of the possible consequences of its recreational use [17]. Nevertheless, numerous reviews highlight the relationship between cannabis use and the development of psychiatric disorders such as psychosis, affective disorders and anxiety disorders [19,20,21,22].

The comorbidity of an addictive disorder with another psychiatric disorder causes a worse prognosis and greater difficulty in treatment [23,24]. This fact has led in recent years to the development of research in the field of dual pathology (DP). We can define DP as the coexistence of a substance use disorder (SUD) and a mental disorder in the same individual in a specific period of time [25]. There is evidence of a greater development of DP in women; however, women are less likely to go to specialized centers for help and, overall, have less social support, which leads them to show less adherence to treatment.

Although it is known that the prevalence of some disorders is higher in men than in women and vice versa, the reasons for these gender and sex differences have yet to be established. It is important to verify these differences and investigate its possible causes to develop adequate and effective prevention and treatment for both men and women. Thus, the present review aims to survey our current state of knowledge about the gender differences in the development of psychotic, depressive and anxious symptoms associated with cannabis use and mention possible explanations for these gender differences.

## 2. Cannabis

Cannabis is a natural drug that humans have been consuming for over 4000 years for medicinal, industrial and ritual purposes [26]. More than 400 chemical compounds can be found in the cannabis plant, of which at least 144 are cannabinoids. Among the cannabinoid compounds, the most important one is tetrahydrocannabinol or delta-9-tetrahydrocannabinol (THC or Δ9-THC), the main psychoactive component of cannabis [27]. Recently, the increase in its use, both medicinal and recreational, its progressive legalization and the change in the cannabis market have caused a greater interest in the research of this drug.

### 2.1. Epidemiology and Risk Factors for Cannabis Use

The most consumed illegal psychoactive substance in recent years is cannabis, with an upward trend observed mainly among the young population [17,28]. It is estimated that 27.2% of the European population aged between 15 and 64 years have consumed it at some time in their lives [29], and this number is as high as 37.5% in countries such as Spain [16]. It is the most consumed drug worldwide, with an increase in the global number of past-year cannabis users of 18% between the years 2010 and 2019 [18]. It is a drug preferentially used by young people. Approximately 17.3% of European students between the ages of 15 and 16 have used it in the last year. Surveys of the general population reported that about 1.8% of people aged 15–64 in the European Union are daily or almost daily cannabis users, having used the drug for 20 days or more in the last month, most of whom (61%) are under 35 years of age [18]. In Spain, the population aged 15–64 that admits to having consumed cannabis daily during the last month reached 2.9%, and 1.9% have a risky use, rating four or more on the Cannabis Abuse Screening Test [16].

Clearly, the perception of risk associated with cannabis use has declined among the young population [16,30,31]. A perception of lower risk is related to higher drug use, although it is still being discussed as to what extent this reduced perception of harmfulness is responsible for the long-term increase in cannabis use [28]. The legalization of cannabis for recreational use is considered to be a main cause of this decreased risk perception. However, the findings of studies that examined the relationship between policy changes and the prevalence of adolescent cannabis use are conflicting. Some studies have found that the legalization of cannabis has increased the prevalence of cannabis use disorder among young people, but others have failed to find any relationship or even suggest a paradoxical decline in use after legalization [32].

On the other hand, it is evident that cannabis is the drug with the highest perceived availability [17], with 59.4% of the Spanish population believing that it would be easy for them to obtain it within 24 h [16]. This possibly also contributes to the fact that, along with alcohol, it is among the first drugs of abuse to be consumed among adolescents. Although the average age of initiation of cannabis use has remained stable since the year 2000, it is below 17 years of age [16,18]. It is well known that adolescence is a critical period of brain development characterized by significant structural change in the brain, specifically in the cerebral cortex [33]. Therefore, this period is also associated with a vulnerability to psychopathology. It has been well documented that a marked upsurge in psychiatric illness, including anxiety, depression and psychotic disorders, occurs during adolescence [34,35]. Recently, it has been demonstrated that the endocannabinoid system plays a role in the development of other systems, in particular dopamine circuits, during adolescence [36].

Currently, the cannabis market is changing, with a very high increase in THC levels in products [17], which are prepared by extracting cannabinoids from the plant to make a product with THC concentrations that usually range from 52 to 69% THC, but which can be as high as 90–95% [37]. This high THC content in cannabis products now available in Europe has been linked to the increase in the overall number of first-time admissions to treatment for cannabis problems in recent years [18]. With respect to the number of admissions to treatment of all entrants, cannabis is at 28.1% in Spain and 36% in the European Union, which increases to 46.8% in treatment demands for first-time entrants [18]. It is also important to highlight that 95.2% of people under 18 who were treated for illicit drug use in Spain did so for problems associated with cannabis use [16].

In addition, the presence of cannabis has continued to increase in Spain in hospital emergencies related to non-medical drug use and in toxicological analyses of deaths due to acute reactions to psychoactive substances. In 2019, more than 50% of hospital medical emergencies related to drug use were due to cannabis; and this drug was detected in 25.5% of those who died of an acute drug reaction, the highest value recorded in the historical series [16]. As it is usually consumed in combination with other substances, such as hypnosedatives, opioids, cocaine or alcohol, it cannot be established how it contributed to the subject’s death. Thus, cannabis abuse, by itself, does not seem to produce overdose like other drugs, but it does carry a significant psychiatric burden [27]. Moreover, cannabis use-related harms are not limited to the mental health domain, but also significantly affect the field of social security (mainly motor vehicle collisions, violence and suicidal behavior), although it has not yet been established what can be considered to be risky use of cannabis [38].

### 2.2. The Endocannabinoid System

Cannabis is the generic name used to refer to psychoactive substances obtained from the Cannabis sativa female plant, known as cannabinoids [39]. Overall, a cannabinoid is an organic compound belonging to the group of terpenophenolics, which can activate the cannabinoid receptors of humans. We can distinguish three general types of cannabinoids: endogenous cannabinoids or endocannabinoids, produced by the human body and which constitute the endocannabinoid system; herbal cannabinoids or phytocannabinoids, naturally synthesized from the cannabis plant; and synthetic cannabinoids, which are similar compounds generated in the laboratory [40]. In general, cannabinoids exert their psychoactive action by binding to brain cannabinoid receptors.

Among the phytocannabinoids, THC, or Δ9-THC, is the cannabinoid with the best known effect and the main psychoactive component of the cannabis plant. Other cannabinoids present in the Cannabis sativa plant with a lower psychoactive potency are cannabidiol (CBD) and delta-8-tetrahydrocannabinol. In particular, CBD is a cannabinoid that is not currently considered to be psychoactive or, at least, not addictive [27,39,41]. In fact, while THC acutely impairs learning and can produce psychosis-like effects and increase anxiety, CBD can enhance learning and seems to have antipsychotic and anti-anxiety properties in humans [41]. Thus, these two compounds appear to have a variety of opposing effects on the brain and human behavior [42], which is probably as result of their different actions on the endocannabinoid system. While THC acts as an agonist in the cannabinoid receptors, CBD seems to have a more antagonist effect, although its mechanisms of action are still not completely understood [41]. Consequently, taken together, CBD can improve the negative effects of THC. However, the market for cannabis has evolved over the past two decades and the THC content in street cannabis has increased dramatically, while its CBD content has remained stable [43] or has decreased to negligible levels [42].

The chemical structure of THC was discovered in 1964, but it was not until 1988 that the first specific brain receptor where cannabinoids act was identified: the so-called CB1 cannabinoid receptor [44]. Similar to what happened with endogenous opiates, while investigating the mechanism of cannabis action, it was discovered that the brain produces chemical substances with a structure similar to that of THC. Therefore, this neurotransmitter system was named endocannabinoid.

The endocannabinoid system is composed of ligands or neurotransmitters, endocannabinoids (anandamide and 2-arachidonoylglycerol), specific receptors (CB1 and CB2) and another biological signal, such as transporters in charge of neurotransmitter receptors and enzymes in charge of neurotransmitter degradation [44].

Endogenous cannabinoids are small lipid molecules that are not restricted to the central nervous system, as they have been detected throughout the body. Moreover, endocannabinoid production in the central nervous system does not only happen in neurons, but also in glial cells [45]. The best known endocannabinoid or endogenous cannabinoid is anandamide, which was identified in 1992. It acts mainly at the brain level on presynaptic receptors regulating the release of other neurotransmitters, such as GABA, glutamate or acetylcholine, in different brain areas, such as the hippocampus, the cerebellum or the thalamus. Anandamide has many actions, such as stimulating hunger or decreasing motor activity, body temperature and pain sensitivity. It also acts outside the central nervous system in organs such as the spleen, where it regulates the immune system. THC simulates the action of anandamide, but with greater intensity and duration [45].

2-Arachidonyl glycerol, or 2-AG, is another endocannabinoid that was identified only three years later, in 1995. It also acts by regulating neurotransmitter release in GABAergic and glutamatergic neurons, being much more abundant than anandamide in the central nervous system, for which it is considered to be the main neurotransmitter of the endocannabinoid system. There are other endocannabinoids, such as virodhamine, arachidonyl-glyceryl-ether, N-arachidonoyl-dopamine and oleamide, among others [45].

Two types of cannabinoid receptors have been documented, CB1 and CB2, which are metabotropic G protein-coupled receptors. However, new families of receptors to which endocannabinoids can bind are being described [44,45]. The CB1 receptor is the most abundant and widely distributed receptor in the central nervous system, both in neurons and glia. However, although the CB2 receptor was initially thought to be found exclusively at the peripheral level, mainly in the immune system, it is now known to be widely expressed in microglia, astrocytes, oligodendrocytes and other brain cells, including neurons. Thus, both cannabinoid receptors of a very complex structure are present in many structures of the nervous system, both central and peripheral, and are linked to many other neurotransmitter systems. In addition, there is evidence of mitochondrial CB1 receptors being involved in the regulation of metabolic processes and memory [45]. Currently, the presence of hemoglobin-derived peptide cannabinoids has been identified in the brain and other organs of the body, suggesting that they are the novel ligands of endogenous CB receptors. These peptide endocannabinoids not only coexist with lipid endocannabinoids, but interact with them. Their peptide nature, i.e., as hydrophilic molecules, provides new biological properties to the cannabinoid signal [44,45].

In summary, as far as is known to date, the main physiological function of endocannabinoids at the brain level consists of the regulation of neurotransmitter release, although they are also primarily involved in synaptic plasticity processes. Both anandamide and 2-AG are produced and released by postsynaptic neurons at the synapse, acting as retrograde messengers on presynaptic receptors. However, anandamide can also be synthesized at the presynaptic level, mediating a postsynaptic form of synaptic plasticity through AMPA receptors in structures such as the nucleus accumbens and hippocampus [45].

It is now known that the endocannabinoid system has many actions, modulating many systems in addition to the nervous system, such as the cardiovascular, immune and endocrine systems. Thus, it is involved in important physiological and psychological processes, including learning and memory, motivation, emotional control, decision making, the regulation of voluntary and learned movements, motor control and spatial coordination, reinforcement, anxiety and stress, the control of the sleep/wake cycle, fear, nociception, eating behavior, as well as neural development, energy metabolism and synaptic plasticity processes [26,44,45].

### 2.3. The Effects of Cannabis Use

Acute cannabinoid use induces changes in brain neurochemistry, such as an increased dopamine release, reduced glutamatergic transmission, the release of endogenous opioids and the inhibition of acetylcholine secretion [39]. These brain biochemical changes are responsible for the acute effects of cannabis, which can be divided into physiological, psychological or behavioral and cognitive effects [42,46].

Among the many physiological effects it produces, we can highlight an increased appetite, anti-emesis, dry mouth, analgesia, drowsiness, sedation, dizziness, decreased intraocular pressure, eye reddening and hypothermia. It also produces neuroendocrine effects, among which we can highlight the activation of the hypothalamic–pituitary–adrenal axis or a decrease in growth hormones, gonadotropin and prolactin. Moreover, it causes effects at the cardiovascular level, such as increased heart rate, cardiac output and blood pressure. In addition, when smoked, which is the main route of consumption, it causes the inflammation of the respiratory tract similar to the effect of tobacco [39]. Thus, despite its many actions, cannabis use has not been associated with a direct risk of death by overdose [26,27]; however, this fact does not imply that it does not have negative consequences.

Relaxation, well-being or euphoria are the main psychological effects of cannabis sought in its recreational consumption, although it can also cause stress and anxiety reactions, especially at high doses. Regarding behavioral effects, we find perceptive alterations and the deterioration of psychomotor performance, such as ataxia, catalepsy and immobility [26,39,46].

Finally, at the cognitive level, cannabis produces a clear attentional deterioration, difficulties in concentration and inhibitory processes, alterations in judgment and working memory, as well as short-term memory, mainly of the verbal type [39,42,46].

In general, we can separate the acute effects of cannabinoids into positive and negative. Within the positive ones, we highlight euphoria, relaxation and sensory intensification. Adverse or negative effects include anxiety, paranoia, impaired psychomotor performance and cognitive dysfunction [42,46]. The increase in THC concentrations in the cannabis consumed is causing an increase in the number of emergencies derived from its acute use, mainly due to panic attacks [26].

It is well known that the repeated use of a psychoactive substance causes adaptive changes in the central nervous system, which means that the effects of the drug after chronic use may be different from acute use. Thus, regular cannabis users present brain alterations, mainly in the regions with the greatest presence of CB1 receptors, such as the prefrontal and limbic areas. For this reason, the continuous consumption of cannabinoids sometimes produces effects that are the opposite of those caused by acute consumption [39]. These changes can be more or less long-lasting, depending on the moment in the life cycle in which the consumption occurs. Thus, exposure to a drug during a critical period of brain development, such as gestation, childhood or adolescence, will have more serious and lasting consequences on neurochemistry and brain anatomy [47,48].

The endocannabinoid system appears in the early stages of fetal development and is involved in the processes that develop and establish synapses, as well as neurogenesis and neuronal differentiation. These processes continue during adolescence, facilitating neurodevelopment thanks to its involvement in neuroplasticity and synaptic function [36,47,49]. In addition, it is related to the regulation of other neurotransmitter systems, such as the glutamatergic, dopaminergic or serotonergic systems, which are all also affected by chronic cannabis use [27,49]. This is why brain differences have been found due to prenatal exposure and chronic cannabis use compared to non-consumers [39,50]. Of note are the alterations observed in the activity of various brain areas, such as the prefrontal cortex, the mesolimbic system, the amygdala, the striatum and the hypothalamic–pituitary axis [27,47,50]. There is also a reduction in the hippocampus and the density of gray matter in different brain regions related to the limbic system [39], and structures involved in executive functions, emotional regulation and reinforcement systems [50,51]. Recently, a longitudinal study demonstrated that cannabis use during middle to late adolescence was related with cortical thinning in a dose-dependent manner, such that greater use from baseline to 5-year follow-up was associated with increased rates of cortical thinning in predominantly prefrontal regions during that same period [47].

Thus, at the cognitive level, high cannabis consumption is related to the impairment of nonverbal learning and episodic memory, not only in the short term but also in the long term [40]. Attentional and impulse control has also been impaired by cannabis abuse, although it seems to improve with abstinence [46,52]. The impairment of executive functions is related to a greater extent with a younger age of onset, a shorter time of abstinence, a higher dosage and a higher THC:CBD ratio [53].

Despite the multiple detrimental effects of continued cannabis use, some studies have demonstrated its usefulness in the treatment of palliative patients [39]. Some cannabinoids, such as cannabidiol in particular, have also been shown to be effective in the treatment of some childhood epilepsies [54], or even in the treatment of some types of schizophrenia [55]. However, a common mistake is to confuse the therapeutic use of certain cannabinoids, such as cannabidiol or medical cannabis, with recreational cannabis use. This confusion often occurs in public discourse and leads to a possible trivialization of the harm that cannabis use can produce in adolescent users, as well as to the reinforcement of the idea that recreational use is a harmless activity [56].

### 2.4. Cannabis Use Disorder

For many years, cannabis was not considered to be addictive. Addiction is defined as a chronic disease, with multiple relapses, characterized by active drug-seeking behavior and a compulsion to use the drug despite harmful consequences. It is considered to be a disease that presents long-lasting alterations of the brain, both in its structure and in its functioning [57]. Today, there is no longer any doubt that the continued use of cannabis can lead to the development of an addictive disorder [22,26,42,51,53]. Despite being a term still used by professionals, the fifth edition of the *Diagnostic and Statistical Manual of Mental Disorders* (DSM-5) [58] does not consider the word addiction to be a diagnostic term, but encompasses it in the definition of “Substance Use Disorder”, which includes from mild to severe states of consumption. Specifically, cannabis use disorder (CUD) is defined by the DSM-5 as “a problematic pattern of cannabis use leading to clinically significant impairment or distress, as manifested by at least two of the following (11 criteria), occurring within a 12-month period” [58] (p. 509).

The factor that has been most closely related to the development of a CUD in cannabis users is the person’s age at the onset of use. Late adolescence and early youth, when the brain has not yet reached full maturity, are the periods of brain development that have shown greater vulnerability in manifesting psychiatric pathology [27,36]. However, other factors, such as the frequency and severity of cannabis use, time of use and periods of abstinence, as well as the presence of comorbid disorders, use of other substances, gender and genetics, influence the severity of CUD.

Neuroimaging studies have shown that subjects with a CUD present a down-regulation of CB1 receptors after a short period of cannabis abuse. These changes are mainly observed in the neocortex and limbic cortex, structures that regulate cognition and emotions, as well as in the ventral striatum, which is involved in motivation and reinforcement [27,51]. Another neurotransmitter system that is highly involved in motivational processes and that is affected in habitual cannabis users is the dopaminergic system. As with any other drug that produces dependence, THC increases the release of dopamine in the ventral striatum after acute use. However, after continued use, there is a reduction in dopamine availability [27,51], similar to that observed in other drug use disorders, but not a decrease in dopaminergic D2/D3 receptors [27], which is also characteristic of addiction to other drugs [57]. Similarly, the glutamatergic system, regulated by CB1 receptors and highly involved in decision making, memory and pathologies such as schizophrenia, is compromised in individuals with a CUD [27,51]. Magnetic resonance imaging studies have revealed, in turn, structural alterations consistent with THC abuse in important brain areas, such as the prefrontal cortex, hippocampus and amygdala [27,47,51]. All of these brain changes observed in subjects with a CUD could explain the development of psychiatric symptomatology associated with cannabis use. In fact, repeated use has also been linked to the development of long-term psychiatric disorders [53].

## 3. Dual Diagnoses in Cannabis Users

Numerous studies have linked cannabis use and psychiatric pathology not only to psychosis and schizophrenia, but also to an amotivational state, depression, anxiety, bipolar disorder and personality disorders [21,26,27,39,40,59,60,61,62,63,64,65]. However, given the complexity of controlling variables in human studies, such as dose or composition of cannabis and the consumption of other drugs, among others, it is very difficult to establish causality between cannabinoid use and psychiatric disorders. Thus, it is still under debate whether comorbid disorders are prior to or a consequence of cannabis abuse [27,66]. For this reason, despite the strong evidence in the association between psychopathology and cannabis abuse, the causal relation of this association is still not understood [21].

The coexistence in an individual of an addictive disorder and a mental disorder in a specific period of time is frequent, and is called dual pathology (DP) [25]. Thus, DP occurs in patients with symptoms that fit the criteria for two different psychiatric disorders, one of them being an addiction. DP is underdiagnosed and poorly treated [67]; in fact, it is not officially recognized in the DSM or CIE nomenclature. However, the term dual pathology is similar to other more commonly used terms, such as dual diagnosis, comorbid or co-occurring disorder or psychiatric comorbidity [67]. The prevalence of DP is high, with 65–85% of addicts in treatment that report having another psychiatric disorder [68] and about 45% of psychiatric patients that report having an addiction [69]. Scientific evidence supports the link between disorders present in DP; therefore, access to a single multidisciplinary care model that integrates and coordinates the mental health network and the addiction network is advocated, enabling personalized bio-psycho-social treatments that do not leave any addict unassisted [23,24].

Patients with DP often present a pattern of polydrug use, and cannabis is one of the most commonly used drugs by these patients [70]. The presence of DP in cannabis addicts under treatment is very high too [21,66]. Firstly, review studies have shown a high comorbidity between CUD and other SUDs, mainly with those induced by legal drugs such as alcohol and tobacco, but also with illegal drugs such as cocaine. It is interesting to note that in those studies, cannabis use was prior to that of other substances [21].

Among all the psychiatric pathologies related to cannabis use, psychosis has been the most widely studied. In an international study with volunteers from 11 locations in Europe and Brazil, it was observed that the probability of developing a psychotic syndrome among daily cannabis users was two to three times higher than in non-users, while in those who used high-potency cannabis, it was one to six times higher than in non-users. This study also highlights the positive relationship between the use of cannabis with higher than 10% THC levels and the development of certain types of psychosis [71]. Not all studies are so categorical in their conclusions, as there are other predisposing factors, such as genetics or childhood trauma [72]. However, increasing evidence indicates a high risk of developing psychosis after frequent cannabis use, especially with high THC levels [21,66]. Both natural and synthetic cannabinoid use in young people has been associated with the occurrence of transient and dose-dependent positive and negative schizophrenic symptoms in healthy individuals not at risk for schizophrenia. However, in adolescents who do present such risk factors, they would cause an earlier onset (between 2 and 6 years of age) and a worse prognosis in the development of schizophrenia [20].

In addition, the percentage of subjects with schizophrenia who use cannabis, even with a CUD, is very high; indeed, it is known that this type of consumption use has been related to the recurrence or worsening of symptoms, both positive and negative, which increases severity and relapses, and hinders adherence to therapeutic and pharmacological treatments [66]. Several studies suggest that cannabis abuse affects antipsychotic treatment, specifically by decreasing adherence to such treatments or worsening abidance, which is consistent with clinical studies indicating that cannabis users in antipsychotic treatment have a higher relapse rate and a worse response to treatment [73]. However, there are no studies as of yet that take into account other determinants, such as THC concentration.

The most repeated reasons to justify cannabis misuse are the enhancement of positive feelings, habit and coping with negative feelings [74]. However, continued cannabis use is associated with the long-term development of depressive and anxious symptoms, the very symptoms that the subject is trying to alleviate during initial use. Thus, it seems that cannabis produces a bidirectional effect, since it can alleviate certain anxious symptoms and reinforce positive mood states when consumed less frequently and with low doses, but depressive and anxious symptoms become more relevant when the frequency of consumption and doses are increased [75]. These dose-dependent effects would be produced mainly by the neurochemical brain changes described in chronic users [27,57].

Cannabis use during adolescence has also been associated with an increased risk of developing depression and suicidal behavior in later years, in this case, without the need for prior risk factors [76]. However, it is not only adolescents who are more vulnerable to pathological cannabis use. Studies in users over 50 years of age, a population that begins to show cognitive deficits due to changes in brain plasticity, have shown that cannabis use has a negative impact. Compared to non-users, this population experiences greater psychological distress and suicidal thoughts, as well as a higher rate of opioid use disorder [77].

The results of longitudinal studies that have evaluated the association between depression and cannabis use are mixed [21,66]. It appears that when the onset of use is early and regular, the risk of depression is higher among cannabis users than among non-users. However, the overlap of the symptoms used to diagnose major depressive disorder with symptoms for cannabis withdrawal syndrome makes it difficult to establish the relationship between cannabis use and depression [21]. Recently, the relationship between the endocannabinoid system and major depressive disorder has been highlighted. Studies have revealed that the endocannabinoid system strongly influences the neurotransmission, neuroendocrine and neuroimmune systems, which have been identified to be dysfunctional in depressive patients [64]. Knowing the involvement of the endocannabinoid system in the etiology of a major depressive disorder can help to understand how cannabis use is associated with the development of depression and suicidal behavior.

Similarly, a relationship between cannabis and anxiety has been observed [21,66]. However, while some studies show that cannabis use, especially during adolescence, increases anxiety levels throughout life, other studies find that anxiety increases when cannabis use ceases [66]. It is known that high doses of THC can cause anxiety symptoms, including panic attacks, and so it has been suggested that continued use may exacerbate anxiety disorders [21]. Overall, the role of cannabis in the etiology, prognosis and treatment of anxiety disorders remains unclear and needs further research [21].

## 4. Gender Differences in Psychiatric Disorders

It is a known fact in psychiatry that sex and gender differences exist in mental disorders, but they have not been widely studied, which means that their causes remain mostly unknown [1]. It is well known that women have a higher lifetime prevalence of anxiety disorders [6,9,10] and depression [7,8], while men develop other disorders such as schizophrenia to a greater extent [3,4,5]. However, the differences between men and women are not only in the prevalence and incidence of pathologies, but also in risk factors, symptomatology, course, prognosis and response to treatment [1]. In addition, gender differences are also observed in substance use disorders [11,12,13,15,78,79]. Thus, men have a higher prevalence of using illegal drugs, such as cannabis, while women generally prefer legal drugs, such as hypnosedatives [18]. These differences depend on the other variables, such as age and educational level. For example, these sex differences are lower in the younger population and higher in subjects with lower levels of education [15]. However, women are not only increasing their consumption of addictive substances [18], but also present clinical profiles of prolonged cannabis use that are different from those of men [4,80]. Women show a faster transition from recreational use to compulsive use, i.e., to dependence, which has been called the “telescoping effect”. Additionally, they show higher levels of craving with more relapses and more frequent and severe withdrawal symptoms. They also show different patterns of use than men and a higher prevalence of dual pathology, which means a worse prognosis, especially because they ask for less help and they have less social support during treatment, leading to higher abandonment rates [14]. This makes it all the more important to develop studies aimed at preventing the development of addictive substance use disorders from a gender perspective.

If we focus on cannabis use, all the latest data reaffirm the fact that men abuse cannabis more often than women in all types of consumption, including those who report having used it once in their lives, in the last 12 months, in the last 30 days, as well as on a daily basis [16,17,18]. Traditionally, men have had an earlier onset of use (probably due to a greater opportunity to access the drug), a higher prevalence and a more severe form of use than women, which would explain the higher CUD rate in men and their greater demand for treatment [11,78]. However, these differences in the prevalence of use are decreasing in the younger population [16,18,26,27,29,42], and both sexes have increased the number and proportion of hospitalizations involved with a CUD diagnosis among people aged 18–25 years old.

It is also confirmed that women develop CUD more rapidly than men, possibly because they manifest greater positive effects and a greater desire to consume [11]. Furthermore, when analyzing the proportion of mental disorders among those hospitalized with CUD, women have a higher proportion and more severe pathologies than men [78].

In summary, most studies find that men not only use cannabis more frequently than women [5,81,82,83,84,85,86,87], but also start using it earlier in life [11,16,78,82]. Probably, the higher consumption rate of men could explain their higher prevalence of CUD at a younger age [85] and with higher criteria of abuse and dependence to cannabis than women [88]. However, this trend may be changing given that, although males continue to consume more, the age of onset is becoming equal for both sexes [16]. In fact, a recent study [83] found no gender differences in the age at which cannabis use began, despite the fact that the same group of researchers had observed them in previous studies [82]. Gender differences in the pattern and reasons for cannabis use have also been reported. A 10-year follow-up cohort study showed that adolescent males consumed mostly in groups and in sport-related situations, while females were more prone to be alone and with the aim of reducing stress [89]. The most repeated reasons for cannabis use in both sexes were “to relax” and “to sleep better”, with women referring most often to the former [83,89]. These gender differences have been observed similarly in studies evaluating other drugs [11,14].

## 5. Gender Differences in the Development of Psychotic, Depressive and Anxious Symptoms Associated with Cannabis Use

The interest in incorporating the gender perspective in dual pathology is quite recent [24]. Some reviews have examined the roles that the interaction between gender and cannabis use have on the development of psychosis [4,84]; however, recently, Prieto-Arenas and Díaz [90] have performed a systematic review on clinically based research evidence of gender differences in the development of psychotic, depressive and anxious symptoms associated with cannabis use. That systematic review was performed on the main databases (PubMed and Web of Science) following PRISMA guidelines on clinical studies published until December 2020. The most important findings to date from the reviewed studies on gender differences in the association between cannabis use and the psychiatric symptomatology described below are summarized in Table 1.

Reviewing the human literature that evaluates the association between cannabis use and the development of psychopathologies, some studies show that cannabis use increases the risk of first-episode psychosis and the development of psychosis more in men than in women [4,84,91], a fact that could be attributed to the greater polyconsumption that men perform [82]. However, when the substance consumed is a synthetic cannabinoid, the risk increases in both sexes [92]. We must bear in mind that these designed drugs have higher levels of THC, which has been related to the appearance of psychiatric comorbidity [21,26]. However, gender differences are not so conclusive when it comes to the effects of cannabis on the age of onset of psychosis. While some studies relate the use of cannabis with an earlier start of first-episode psychosis in both sexes, without finding gender differences [5,84,93,94], other studies find that the use of this drug reduces the age of onset of psychosis more in women than in men [4,81,91,95], eliminating the gender differences observed in general in the age of onset of psychotic disorders [93].

Thus, cannabis use, whether recreational or compulsive, has been related to the appearance of symptoms of psychosis in both the adolescent [88,89] and young population [96], and in subjects with risk factors for the development of psychosis [97]. In a nonclinical population of university students, women reported greater intensity of psychotic experiences associated with cannabis use than men [96]. Women with risk factors for the development of psychosis also presented a greater severity of general psychiatric pathology related to the consumption of this drug [97]. However, in this same population, male users exhibited a greater severity of negative psychotic symptoms [97].

On the other hand, women diagnosed with a CUD also showed more psychotic and depressive symptoms than men of the same age range (23–25 years), although there was no relationship between the frequency of cannabis use and the age of onset of symptoms [89]. In addition, it should be noted that after one year of treatment, men with CUD significantly reduced their use of cannabis, while women did not [86].

Among patients with first-episode psychosis, cannabis use has been observed to worsen psychological, social and work activity in men, while the opposite result was surprisingly found in women [83]. This finding could be due to the lower number of women being evaluated in the study; therefore, more in-depth research is needed to fully understand this fact [90].

In contrast, in patients with a cannabis-induced psychotic disorder, it is men who have a greater severity of general psychopathology, in addition to a greater intensity and prevalence of positive symptoms [98], while women show more negative symptoms [86]. This coincides with two reviews in which it is concluded that cannabis use in men increases the manifestation of psychotic symptoms and hospitalizations [4,91]. However, other studies do not find these gender differences in first-episode psychosis patients, finding a similar severity of clinical symptoms and length of hospitalization period in both sexes [3,83]. It should be noted that with the use of synthetic cannabis, the levels of agitation in women with psychotic pathology increase compared to those of men [92].

In summary, we can affirm that all the studies demonstrate the existence of an association between the use of cannabis and the appearance of psychotic symptoms. However, despite gender differences being observed, they are not always confirmed by all studies, depending mainly on the population studied [90]. It seems that female cannabis users would manifest a greater intensity of psychotic symptoms and general psychiatric pathology in both the previously asymptomatic population and people with a problematic use of cannabis [88,89,96,97]. Therefore, cannabis use appears to be a higher risk factor for women than men, and is associated with a worse prognosis of schizophrenia [84] and CUD [86] in women. On the other hand, although male cannabis users with first-episode psychosis showed a worse quality of life [83], there were no gender differences in the severity of clinical symptoms [3,83]. Nonetheless, when cannabis has already induced a psychotic disorder, it is men who show a greater severity of general psychopathology [98].

Cannabis use has generally been associated with depressive symptoms [76]. However, while some studies have shown that women present this cannabis-related symptomatology to a greater extent than men [99], others do not find such differences [95,96]. Therefore, this association should be specified, since some studies focus on gender differences found in the development of depressive symptoms, others in major depression and others in suicidal ideation [90].

The increase in the frequency of cannabis use is predictive of depressive symptoms in adolescents of both sexes [100,101], although more significantly in women [100,101,102], who show greater psychological distress than men [103]. These results coincide with the fact that women present more depression in the general population [7,8]. On the other hand, other studies indicate that cannabis use is a predictor of depressive symptoms with a greater severity only in men [104,105], which is maintained over the years [52]. This would explain why cannabis use has been seen to increase the probability of developing episodes of major depression in males [106]. This notwithstanding, gender differences between cannabis abuse and major depression have not yet been found [103].

Among subjects with cannabis misuse, men show more depressive symptoms at younger ages (19–20 years), while women show higher depressive and somatization symptoms [85,88,102,107,108] at later ages (23–25 years) [89]. Probably, these gender differences in the age of onset of depressive symptoms are due to the fact that men start consumption earlier in life [82].

It has also been described that women with risky cannabis use show a higher possibility of suicide than men in late adolescence [85]. However, men manifest a greater probability of suicidal ideation when increasing the frequency of consumption [95,109], and the gender differences disappear when the sample is extended to the general population [109]. On the other hand, although no relationship has been found between the risk of suicide and the age of onset in the development of CUD in any sex [85], a relationship has been observed between the suicidal history and the onset of cannabis use in women [95]. This is in keeping with the fact that the main reason for consumption reported by women is to relax and reduce stress, probably using cannabis as self-medication [83,89].

Additionally, female cannabis users with first-episode psychosis or diagnosed psychosis have shown greater dysphoria and depression than men [86,97,98]. However, among psychotic males who use cannabis daily, two trends have been found according to age. While it is observed that the increase in the frequency of consumption reduces the probability of suicide attempts in the youngest, when the age range increases (35–64 years), there is a relationship between daily consumption and a greater number of suicide attempts compared to non-users [110]. This change in the direction of the relationship between use and suicide in men could be the result of differences between the acute and chronic effects produced by cannabis [39].

Finally, there are few studies that relate cannabis use to the development of anxious symptomatology, as most studies focus on assessing anxiety as a risk factor for its use. In general, women have the highest levels of anxiety and related disorders among adolescents [89,102], the general population [89,97,99,111], and psychiatric patients [85,98,107,108], with the biggest gender differences found in late adolescents [89]. Specifically, it has been described that the men and women with low stress tolerance are those who show the most problems related to cannabis use, and in particular women who use the drug as a stress-coping mechanism [112], in a manner similar to that previously commented upon [83,89]. Hence, a positive relationship has been found between cannabis abuse and generalized anxiety disorder in women in the general population, while men showed a negative relationship with panic disorder [111]. In addition, women manifest greater anxiety than men in the periods of abstinence [95,113]. As observed with depression, Foster et al. [85] found no relationship between the age of onset of CUD and anxiety problems.

In conclusion, the scientific evidence reveals the existence of gender differences in psychiatric symptoms associated with cannabis use, although the direction of such differences is not always clear [90]. A lack of information in studies about variables such as the THC level in the cannabis used, the frequency of use or the age of onset of cannabis use makes it difficult to know the causes for the conflicting results. Besides, few studies consider the specific characteristics of women diagnosed with dual pathology, although all the data indicate a higher prevalence of drug-associated pathologies and a worse prognosis in women [24]. For this reason, it is necessary to delve deeper into this issue and address gender differences to create more individualized prevention strategies and more effective treatment for dual disorders related with cannabis abuse.

**Table 1 brainsci-12-00388-t001:** The most important findings to date from the human literature on gender differences in the association between cannabis use and the development of psychotic, depressive and anxious symptoms. M: male; F: female; CUD: cannabis use disorder.

Symptom	Population Profile	Results	References
Psychosis	nonclinical	Cannabis use increases the risk of first-episode psychosis and the development of psychosis more in men than in women **(M > F)**	[4,84,91]
		Synthetic cannabis use increases the risk of the development of psychosis both in men and in women **(M = F)**	[92]
		Cannabis use is related with an early start of first-episode psychosis both in men and in women **(M = F)**	[5,84,93,94]
		Cannabis use reduces the age of onset of psychosis more in women than in men **(F > M)**	[4,81,91,95]
		Greater intensity of psychotic experiences are associated with cannabis use in women than men **(F > M)**	[96]
	with risk factors for psychosis	Cannabis use is related with a greater severity of general psychiatric pathology in women than men **(F > M)** and with a greater severity of negative psychotic symptoms in men than women **(M > F)**	[97]
	with CUD	Women present more psychotic symptoms than men **(F > M)**	[89]
		Women present worse responses to treatment with more relapses than men **(F > M)**	[86]
	with first-episode psychosis	Cannabis use is related with worsening psychological, social and work activity in men than women **(M > F)**	[83]
		Cannabis use is related with a severity of clinical symptoms and length of hospitalization period similar in both sexes **(M = F)**	[3,83,92]
	with a cannabis-induced psychotic disorder	Men present a greater intensity and prevalence of positive symptoms than women **(M > F)**	[4,91,98]
		Women present a greater intensity and prevalence of negative symptoms than men **(F > M)**	[86]
Depression	nonclinical	Cannabis use is predictive of depressive symptoms both in men and women **(F = M)**	[95,96]
		Cannabis use is predictive of depressive symptoms more in women than men **(F > M)**	[99,100,101,102,103]
		Cannabis use is predictive of depressive symptoms with a great severity in men **(M > F)**	[52,104,105]
		Cannabis use increases the development of major depression in men **(M > F)**	[106]
		Cannabis use is related with suicidal ideation both men and women **(M = F)**	[109]
	with a cannabis misuse	Women show more depressive and somatization symptoms than men **(F > M)**	[85,88,102,107,108]
		Men show more depressive symptoms at younger ages **(M > F)**, while women do so at later ages **(F > M)**	[89]
		Women show a higher probability of suicide than men in late adolescent **(F > M)**	[85]
		Men with a high frequency of consumption manifest a greater probability of suicidal ideation than women **(M > F)**	[95,109]
	with psychosis	Women cannabis users present greater dysphoria and depression than men **(F > M)**	[86,97,98]
Anxiety	general	Positive relationship between cannabis abuse and generalized anxiety disorder in women and negative relationship between cannabis abuse and panic disorder in men	[111]
	with CUD	Women present greater anxiety than men during abstinence **(F > M)**	[95,113]

## 6. Possible Explanations for the Gender Differences in Dual Diagnosis Associated with Cannabis Use

When seeking the possible causes of sex and gender differences in psychopathology in general, the complexity of the issue becomes clear, but more so when these differences in disorders are associated with the consumption of a drug. First of all, we must better understand the contributions of biological sex to these differences, from the chromosomal sex-dependent genetic risk to brain and hormonal differences resulting from the process of sexual differentiation, sex-specific epigenetic markers, the different response to stress and the immune response [10,79,114,115], as well as the importance of the role of gender orientation (masculinity and femininity). There is a lot of evidence that social factors, such as the different frequency or pattern of drug consumption [8,9], a higher rate of sexual abuse, greater risk of suffering interpersonal stressors and greater social discrimination in women, are related to increased stigma, social penalties and barriers to access to treatment [15,24,89], which can likely be reasons for the differences between men and women in the development of dual pathology.

Thus, there are many possible causes of these gender differences in the association between cannabis use and the development of psychiatric symptoms and, overall, we can distinguish between those that are due to social aspects and those that are of biological origin. However, only the possible biological causes that have received greater interest in research in recent years will be discussed below.

One of the most studied traditional explanations is the role of gonadal hormones, specifically 17-beta-estradiol, the main form of estrogen. The lower prevalence and later onset of psychosis in females has been mainly explained by the estrogen protective hypothesis [2,84,114]. Ample scientific evidence has related low levels of estrogen with the development or worsening of psychotic symptoms [116]. Concurrently, these gonadal hormones are also linked to sex differences in many other psychiatric disorders [117] and in the neurobehavioral response to drugs of abuse [13]. It is known that cyclic fluctuations of estrogens modulate the mesolimbic reward and that the levels of estrogens enhance addiction vulnerability in females [13,118]. Therefore, female gonadal hormones have been considered to be critical for sex differences in addiction. Overall, estrogens have been shown to be both protective in the development of pathologies such as psychosis, and a risk in the development of other pathologies, such as depression, anxiety and drug use disorders. Estradiol promotes neuronal sprouting and myelination, enhances synaptic density and plasticity, facilitates neuronal connectivity, acts as an anti-inflammatory and as an antioxidant agent, inhibits neuronal cell death, may mediate BDNF expression and activity and positively influences mitochondrial function [2]. Both the protective role of estrogens in psychotic symptom expression and its role in the greater vulnerability to drug effects are mainly produced not only through the modulation of sensitivity and number of dopaminergic receptors, but also on other key neurotransmitter systems, such as the glutamatergic, serotonergic, noradrenergic and cholinergic systems [13,116]. Thus, estrogens affect many cognitive processes by altering transmission in various neurotransmitter systems, although its role on dopamine-dependent cognitive processes has been the most widely studied. It is known that estrogens alter dopamine function at various stages in transmission by increasing dopamine receptor density and dopamine availability, both in terms of tonic and phasic levels of dopamine, by binding at dopamine D1 and D2 receptors and by decreasing the affinity of the dopamine transporter [119].

Additionally, the gonadal hormones also have an important role in the sex differences of the stress response [117]. These hormones of the hypothalamic–pituitary–gonadal axis have been relevant to determine sex differences in the adult hypothalamic–pituitary–adrenal (HPA) axis function after stress via their activational and organizational effects. Knowing sex differences in the HPA axis is key to understanding the development of pathologies that are differentially present in men and women. The dysregulation of the HPA axis is a hallmark of many stress-related diseases, such as depression, anxiety, psychosis and drug use disorders, among many other pathologies [115,117]. Females typically present a more robust neuroendocrine response to stress than males; however, besides that, females have lower levels of negative feedback on the HPA axis than males. Thus, while the levels of corticoids and ACTH in response to stressors of different modalities are higher in females than males, the activation of limbic structures, such as the prefrontal cortex, cingulate cortex, piriform cortex and the hippocampus, known to activate inhibitory inputs to the HPA axis, is lower in females. Numerous studies reveal that estrogens enhance the activity of the neuroendocrine response, directly and indirectly, in HPA function; however, androgens generally inhibit the activity of this axis. These sex differences of the axis are also due to the organizational effects of gonadal hormones [117,120,121,122]. Nevertheless, there are still many unresolved questions regarding the sex differences in the long-term effects of chronic and early stress for completely understanding the role of stress on the development of psychiatric pathology in women and men [117] and the effect of the endocannabinoid system on these sex differences in the development of vulnerable phenotypes to the appearance of psychiatric disease [123].

Finally, the fluctuations of these female gonadal hormones also appear to be responsible, in part, for the existence of the sexual differences in the endocannabinoid system (ECS). Sex differences in the expression and function of the endocannabinoid system in the brains of rodents have been found in several studies (for review, see [124]). Thus, females presented greater levels of 2-AG and anandamide in the pituitary glands than males [125,126]. Additionally, CB1 receptor affinity was greater in females than males in amygdala [127], in the striatum and in the limbic forebrain [121]. In contrast, CB1 receptor protein density was greater in males than females in the anterior cingulate [128], the prefrontal cortex [128], the hippocampus [129], the hypothalamus [127] and the midbrain [121]. However, CB1 receptor protein density in the amygdala was greater in males than females, increasing with ovariectomy in one study [128], and greater in females than males in another study [127]; moreover, CB1 receptor protein density was not found to be different between males and females in the hypothalamus, striatum and limbic forebrain [121]. The expression of CB1 receptor protein or mRNA was greater in male than female rats in the anterior pituitary [126], the mesencephalon [121] and the prefrontal cortex and the amygdala [128]. Furthermore, cannabinoid receptor distribution in humans has also been studied; however, differences in the methodologies used in the few studies performed to date make it difficult to obtain any definitive conclusions [124]. When utilizing positron emission tomography to examine cannabinoid receptor densities, female cannabis non-users presented greater CB1 receptor density than men [130,131]. On the contrary, greater CB1 receptor density in men versus women among healthy non-user subjects was reported too [132]. Nevertheless, sex hormones, particularly estrogens, have been observed to modify cannabinoid receptor expression and the affinity and efficacy of cannabinoid ligands differently in male and female; however, menstrual phase was not reported in all studies [124]. Hence, the existence of conflicting results found both in animal models and human studies support the need for further research about the sex differences in endocannabinoid system.

These sex dimorphisms described in cannabinoid receptors in certain brain areas, mainly in the limbic forebrain, hypothalamus, hippocampus and amygdala, began to be assessed a few years ago in an attempt to explain the sex-dependent differences in the observed cannabinoid effects [95,99,121,122,123]. Accordingly, the growing research in the field of sex differences in ECS has opened up a promising perspective to find the possible reasons behind the gender differences in dual diagnosis associated with cannabis use [133,134]. The ECS has proven to be essential in the developing nervous system and, in the mature nervous system, it modulates the neuronal activity and network function of a great number of physiological, cognitive, behavioral and emotional processes, as discussed above [135]. Therefore, when the dysregulation of the ECS occurs, psychopathologies might arise, as has already been recognized in diseases such as depression [64].

In addition, there is scientific evidence showing that the repeated use of cannabis produces changes in brain structures and function, not only in the ECS itself, but also in other systems, such as the dopaminergic, glutamatergic and GABAergic neurotransmission systems (for review, see [27]). These detected alterations are also associated with a higher risk of developing psychiatric pathologies, and they may be much more durable over time if consumption has occurred during a critical period of brain development, such as adolescence [133]. However, there are studies that have demonstrated that adolescent cannabinoid exposure has sex-specific effects on CB1 receptor density, on other receptor systems, such as glutamate, GABA and dopaminergic systems, and on brain structures and functions, such as the striatum, neocortex and cerebellum (for review, see [95]). Thus, one of the main explanations for the greater dual diagnosis in female cannabinoid users may be the sex differences found in the consequences of adolescent cannabis exposure [133]. Nonetheless, as of yet, there are few human studies assessing consistent differences in the effects of cannabis on brain structures and functions between men and women. In conclusion, it is clear that men and women differ in the long-term repercussions of cannabis use on their health [2], but more research is needed to determine the causes of these sex differences.

## 7. Conclusions

In recent years, cannabis use has been increasing in a very important way among the youngest [16,17,18,28,29], likely due to a misleading perception of low risk [16,30,31]. The increase in THC concentrations in cannabis products [17], along with the fact that the drug begins to be consumed in the critical period of brain development that is adolescence [16], augur an import increase in the prevalence of CUD and DP in the coming years. In addition, the higher percentage of young women using cannabis, which is on a par with men, and their demonstrated greater vulnerability to develop drug misuses [11], seem to predict an increase in health problems associated with cannabis use in the female population. However, most prevention and treatment programs are aimed at men, the largest consumers to date [24]. We must bear in mind that interest in research on gender differences is quite recent, and to consider the gender perspective in studies is not only to include men and women as experimental subjects, but also to analyze the gender differences, their possible causes and the implications for the diagnosis and treatment of such differences. Only in this way will it be possible to understand the variations by sex/gender in the ratios of prevalence, comorbidity, severity and chronicity of psychopathologies, improving knowledge about women’s mental health. Therefore, we believe that reviews like the present one contribute to demonstrate and to highlight the gender differences in mental disorders more concretely in dual diagnosis. The results of the revised clinical studies clearly show the existence of gender differences in psychiatric symptoms associated with cannabis use [4,52,81,83,84,85,86,88,89,90,91,95,96,97,98,99,100,101,102,103,104,105,106,107,108,109,111,113]. Although these results are not conclusive, they seem to indicate a higher vulnerability of women in the development of psychosis and anxiety, while men seem to be more vulnerable to developing depressive symptoms with long-term cannabis misuses. However, the few studies conducted and the heterogeneity of the methodologies employed make it difficult to obtain clear conclusions in the direction of gender differences. Studies should control important variables such as the frequency of drug use and THC concentrations in cannabis consumed, among others. Moreover, possible explanations for the gender differences in dual diagnoses associated with cannabis use are abundant, with many complex interactions. We must consider the role of estrogens and its relationship with the HPA axis for the sex differences in the neuroendocrine stress response, a key factor in the development of many psychiatric pathologies [115,117]. Most recently, the study of sex differences in the ECS seems to be a key approach to discover the possible reasons behind the gender differences in the dual diagnosis associated with cannabis use. The growing knowledge of the role of this neurotransmission system in the development of the brain and in the function of others important neurotransmission systems, such as the dopaminergic, glutamatergic and GABAergic systems, suggest that its alteration by cannabis use during adolescence may be key to understanding gender differences in DP. Hence, future research in this field is necessary to be able to discern more clearly the gender differences found in the association between cannabis use and the development of psychiatric symptoms, as well as their possible causes.

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
