# Peer review of "Gender Differences in Dual Diagnoses Associated with Cannabis Use: A Review"

_brainsci, 2022, doi:10.3390/brainsci12030388_

Round 1
Reviewer 1 Report
Thank you for very informative article. I reviewed your work closely and I have few suggestions.
- I suggest writing "dual diagnoses" instead of "dual diagnosis" in the title as article is about multiple psychiatric disorders not just one, that's why use plural form of diagnosis.
- Line 86 - did you mean to write risk use or risky use? I addiction literature risky use is an acceptable term.
- Line 138 - "Human organism" could be switch to human or humans
- Line 360 - please change the range from 3 to 2 to 2 to 3.
- Line 485- You are writing University female. Did you mean to write Universal?
- Line 501- Change "psychopathy" to "psychopathology"
- Line 541 - Consider changing "risk cannabis use" to "risky cannabis use"
Author Response
REVIEWER 1:
- Reviewer says that “I suggest writing "dual diagnoses" instead of "dual diagnosis" in the title as article is about multiple psychiatric disorders not just one, that's why use plural form of diagnosis.” Following the Reviewer’s suggestion, we have changed “dual diagnosis” for “dual diagnoses” in the title of the paper and in the third subtitle (line 329).
- Reviewer says that “Line 86 - did you mean to write risk use or risky use? I addiction literature risky use is an acceptable term.” Following the Reviewer’s suggestion, we have changed “risk use” for “risky use” in line 86.
- Reviewer says that “Line 138 - "Human organism" could be switch to human or humans.” Following the Reviewer’s comment, we have changed “the human organism” for “humans” in line 138.
- Reviewer says that “Line 360 - please change the range from 3 to 2 to 2 to 3.” We have made this change.
- Reviewer says that “Line 485- You are writing University female. Did you mean to write Universal?” The cited study [94] evaluated a nonclinical population, specifically university students. We have clarified the sentence, which now reads as follows: “In a nonclinical population of university students, women reported greater intensity of psychotic experiences associated with cannabis use than men [94].”
- Reviewer says that “Line 501- Change "psychopathy" to "psychopathology".” We have made this change.
- Reviewer says that “Line 541 - Consider changing "risk cannabis use" to "risky cannabis use".” We have made this change.

Reviewer 2 Report
It is a very good submission. From my point of view the word "review" should appear at the title of the paper.
The abstract should reflect the discussion and conclusion sections in a clearer and more explicit way.
Some evidence about dual schizophrenia and this condition being always associated to cannabis use, even when a whole sample of oatients with dual diagnosis is described, should be said (e.g. Serrano-Serrano, A.B. et al. 2021; Marquez-Arrico, J.E., et al 2019).
Although this is not a systematic review, the method and research strategies must be explained. Moreover, the reason for not developing a systematic review work should be established. In any case, ked words, criteria and the mehtod of analysis of the publications must be indicated.
Author Response
REVIEWER 2:
- Reviewer says: “From my point of view the word "review" should appear at the title of the paper.” We agree with the Reviewer’s suggestion, so the title now reads as follows: “Gender differences in dual diagnoses associated with cannabis use: a review”.
- Reviewer says: “The abstract should reflect the discussion and conclusion sections in a clearer and more explicit way.” Following the Reviewer’s comment, we have changed the Abstract in order to clarify the discussion and conclusions sections. The abstract section now reads as follows: “… The present work analyses the literature to determine the existence of gender differences in the development of psychotic, depressive and anxious symptoms associated with cannabis use. First, we describe cannabis misuse and its consequences, paying special attention to adolescent subjects. Second, the main gender differences in psychiatric disorders, such as psychosis, depression, anxiety and cannabis use disorders, are related. Subsequently, we discuss the studies that have evaluated gender differences in the association between cannabis use and the appearance of psychotic, depressive and anxious symptoms; and we consider the possible explanations for the identified gender differences. In conclusion, the studies referred to in this review reveal the existence of gender differences in psychiatric symptoms associated with cannabis use, although the direction of such differences is not always clear.”
- Reviewer says: “Some evidence about dual schizophrenia and this condition being always associated to cannabis use, even when a whole sample of patients with dual diagnosis is described, should be said (e.g. Serrano-Serrano, A.B. et al. 2021; Marquez-Arrico, J.E., et al 2019).” Following the Reviewer’s suggestion, we have added this date in section 3 (lines 352-3): “Patients with DP often present a pattern of polydrug use, and cannabis is one of the most commonly used drugs by these patients [Serrano-Serrano et al 2021].”
- Reviewer says: “Although this is not a systematic review, the method and research strategies must be explained. Moreover, the reason for not developing a systematic review work should be established. In any case, ked words, criteria and the mehtod of analysis of the publications must be indicated.” The present study is a thematic review, which intends to analyse the literature about the existence of gender differences in the development of dual diagnoses associated with cannabis use, mainly in the development of psychotic, depressive and anxious symptoms, since these are the pathologies more often related with cannabis use. Recently, two authors have made a systematic review about clinical-based research evidence of gender differences in the development of psychotic, depressive and anxious symptoms associated with cannabis use, which is under review in Adiciones journal. A systematic review was performed in the main databases (Pubmed and Web of Science) following PRISMA guidelines on clinical studies published until December 2020 using the combination of terms “cannabi* OR marijuana OR marihuana” AND “women OR female OR sex OR gender OR sex differences OR gender differences” AND “psychiatric disorder* OR patholog* OR psychosis OR schizophreni* OR depressi* OR anxiety”. The results of this systematic review together with studies published from January 2021 are discussed in Section 5 of the present study. This information has been added in Section 5 of the manuscript: “recently, Prieto-Arenas and Díaz [under review] have performed a systematic review about clinical-based research evidence of gender differences in the development of psychotic, depressive and anxious symptoms associated with cannabis use. That systematic review was performed in the main databases (Pubmed and Web of Science) following PRISMA guidelines on clinical studies published until December 2020.”

Reviewer 3 Report
In this review, the authors investigate the gender differences in dual diagnosis associated with cannabis use, which is an interesting subject. Clearly understanding the gender differences underlying the psychiatric adverse events which appear in cannabis users could be useful in treating these patients. However, the manuscript needs improvement to be relevant.
However, the manuscript needs improvement to be relevant. At this point, it does not offer concrete data.
Major changes
- Section 5 lacks any concrete data! Please insert a table (or more, depending on the number of studies you find) containing all conclusive clinical/observational studies reporting psychiatric adverse effects. Please mention the type of adverse reaction and their frequency in the population of the study (woman vs. men, when possible).
- Section 6
It seems that the main differences are not emphasized enough. You should clearly mention the
mechanisms underlying sex differences in dual diagnosis – it seems you refer to (even though it is not clearly stated):
- Hormonal influences
- HPA axis-endocannabinoid interaction
- Sexual dimorphism of the endocannabinoid system
Line 626-627
“Sex dimorphisms in cannabinoid receptor density and affinity in certain brain areas, such as the limbic forebrain, hypothalamus, hippocampus and amygdala, have been demonstrated both in animal models and human studies”
! PLEASE, expand this as it is important for the section.
- The various sections of your review lack a conclusion. Each section should be ended with a short conclusion.
- Section 7 must be rewritten. You do mention the difficulties that prevent drawing clear conclusions, which is good but clearly state the differences observed by reviewing all data.
Minor changes
Please change title of section “2.2 How cannabis works in our brain” – Indeed, THC&others activate cannabinoid receptors, but you talk very much about the physiological effect of the endocannabinoid system, not on the results of THC activating those receptors. Maybe you should have two separate sections about these two different things? Or at least, clearly delimit them within the section.
Lines 21-22
“the direction of such differences are not always clear” – lacks subject-verb agreement
Line 167
“biological signal termination processes such as transporters in charge of neurotransmitter receptors and enzymes in charge of neurotransmitter degradation”. Please rephrase for better understanding
Lines 336-338
“For this reason, despite the strong evidence, the causal relation of this association is still not understood” – there is strong evidence or it is not understood…Please rephrase.
Lines 428-429
“These differences depend on other variables such as age and educational level, being lower in the younger population and higher in subjects with lower levels of education” – a word of connection is needed.
Author Response
REVIEWER 3:
Major changes
Following the Reviewer’s suggestion, we have summarized in a table the most important findings until today about gender differences in the association between cannabis use and psychiatric symptomatology described in Section 5.
Besides, we have totally rewritten Sections 6 and 7 following all the commentaries from Review. We believe that these sections have greatly improved after taking into account the suggestions.
Minor changes
- Reviewer says: “Please change title of section “2.2 How cannabis works in our brain” – Indeed, THC&others activate cannabinoid receptors, but you talk very much about the physiological effect of the endocannabinoid system, not on the results of THC activating those receptors. Maybe you should have two separate sections about these two different things? Or at least, clearly delimit them within the section.”
Following the Reviewer’s suggestion, we have changed the title of section 2.2 “How cannabis works in our brain” and it now reads as follows: “The endocannabinoid system”.
- Reviewer says: “Lines 21-22: “the direction of such differences are not always clear” – lacks subject-verb agreement”
The Reviewer is right in that there is a lack of subject-verb agreement, the sentence has been changed and it now reads as follows: “the direction of such differences is not always clear”.
- Reviewer says: “Line 167: “biological signal termination processes such as transporters in charge of neurotransmitter receptors and enzymes in charge of neurotransmitter degradation”. Please rephrase for better understanding”
Following the Reviewer’s suggestion, we have changed the phrase and it now reads as follows: “The endocannabinoid system is composed of ligands or neurotransmitters, endo-cannabinoids (anandamide and 2-arachidonoylglycerol), specific receptors (CB1 and CB2) and another biological signal, such as transporters in charge of neurotransmitter receptors and enzymes in charge of neurotransmitter degradation [44]”.
- Reviewer says: “Lines 336-338: “For this reason, despite the strong evidence, the causal relation of this association is still not understood” – there is strong evidence or it is not understood…Please rephrase.”
There is strong evidence in the association between psychiatric disorders and cannabis abuse; however, it is still not understood if cannabis abuse is the cause of psychiatric disorders or the consequence of the development of such disorders. We have changed the phrase and it now reads as follows: “For this reason, despite the strong evidence in the association between psychopathology and cannabis abuse, the causal relation of this association is still not understood”.
- Reviewer says: “Lines 428-429: “These differences depend on other variables such as age and educational level, being lower in the younger population and higher in subjects with lower levels of education” – a word of connection is needed.”
Following the Reviewer’s suggestion, we have changed the phrase and it now reads as follows: “These differences depend on the other variables such as age and educational level. For example, these sex differences are lower in the younger population and higher in subjects with lower levels of education”

Round 2
Reviewer 2 Report
accept if changes were made.
Reviewer 3 Report
The authors revised the article according to suggestions and the manuscript is greatly improved. I consider there are no further changes needed.